# A Fast Method for Multidimensional Joint Parameter Estimation of Polarization-Sensitive Arrays

**DOI:** 10.3390/s23198193

**Published:** 2023-09-30

**Authors:** Zheqi Zhang, Xiang Lan, Xianpeng Wang

**Affiliations:** School of Information and Communication Engineering, Hainan University, Haikou 570228, China; 21220854000195@hainanu.edu.cn (Z.Z.); wxpeng2016@hainanu.edu.cn (X.W.)

**Keywords:** polarization-sensitive arrays, joint parameter estimation, Newton’s method, threshold function

## Abstract

The paper proposes a fast method for the multidimensional parameter estimation of a polarization-sensitive array. Compared with conventional methods (e.g., MUSIC algorithm), the proposed method applies an iterative approach based on Newton’s method to obtain joint estimation results instead of a spectral search and dimension reduction. It also extends the original Newton method to the 4D scale using the Hessian matrix. To reduce the complexity of establishing the aim function, Nystrom’s method is applied to process the covariance matrix. A new threshold is also proposed to select the results, which can accomplish the parameter estimation with a small number of iterations while guaranteeing a high estimation accuracy. Finally, the proposed algorithm is analyzed in detail and the numerical simulations of various algorithms are compared to verify its effectiveness.

## 1. Introduction

In the field of array signal processing, the direction of the array has been extensively researched and developed over decades, and is widely used in radar, sonar, medical and other fields. After a long time of research, scalar sensor arrays have achieved an accurate estimation of signal parameters by various methods. The most classic of these is the MUSIC (multiple signal classification) algorithm. For example, in the article [1,2], the orthogonality of the signal subspace and noise subspace is used to construct a spatial spectral function, and then the parameters to be measured are estimated by traversing the spatial spectrum. In [3], the ESPRIT (estimating signal parameter via rotational invariance techniques) algorithm based on rotational invariance techniques is applied, which makes full use of the rotational invariance between array elements and improves the speed of the algorithm by avoiding a tedious spectral peak search. However, in addition to the DOA angles θ and ϕ, sometimes, we need more signal parameters to ensure the accuracy and high resolution of the information. In these cases, the traditional scalar array obviously cannot meet our needs.

At this point, EM (electromagnetic) vector sensor arrays were first applied to collect electrical and magnetic information from impact signals. The reason is that the polarization components under the influence of electric and magnetic fields describe the vectorial properties of the EM wave. The estimation of this parameter facilitates the improvement of the system’s resolution and immunity to interference. The MUSIC algorithm, for example, is widely used in dipole antenna element arrays [4,5,6,7]. In [5], the authors investigate the maximum element distance constraint (MISC) theory for sparse vector sensor arrays. The MISC theory is proposed, and the MUSIC algorithm based on the quaternion theory under the vector MISC array is proposed, which takes into account the advantages of the quaternion model, MISC array, and MUSIC, and improves the performance of the DOA and polarization information estimation. In addition to the MUSIC algorithm, the ESPRIT algorithm based on the rotational invariance technique in [8,9,10,11] has been applied to polarization-sensitive arrays. To reduce the computational complexity, a search-free algorithm centered on the ESPRIT algorithm is proposed in [12] to estimate the four-dimensional parameters of the PSFDA-MIMO radar. This algorithm avoids the high-complexity problem of four-dimensional joint parameter estimation. Other subspace-based algorithms are often applied to vector sensor arrays, such as the propagation method (PM) and the parallel factorization (PARAFAC) algorithm [13]. The algorithms listed above can be used to obtain the electromagnetic information of the signal using vector sensors, but the methods used rely on a spectral function search and eigenvalue decomposition; moreover, the longitudinal dimension of the signal covariance matrix is expanded to three times its original size under polarization-sensitive arrays, which brings a great computational cost and complexity.

To avoid this, scholars have applied sparse signal representation (SSR)-like algorithms to the estimation of the DOA and polarization information of polarization vector sensor arrays. For example, in [14], an algorithm based on a sparse representation is proposed to estimate only the 1D dictionary and 2D DOA angles. A novel cross-covariance matrix without polarization components is constructed in the estimation process to decouple the DOA and polarization component information to solve the joint estimation problem. The useful information contained in the received signal is fully utilized, thus improving the estimation accuracy. Most of the above-mentioned multidimensional parameter estimation methods use a joint estimation or dimensionality reduction approach, and they fundamentally avoid the problems of spectral function search and feature decomposition, which can reduce the computational burden while ensuring the estimation accuracy and have more outstanding performance under a low signal-to-noise ratio and fast beat. However, the above algorithms still have some shortcomings. The sparse signal representation class of algorithms relies on the division of the grid, and off-grid error estimation is required when the results to be measured do not fall on the divided grid. This makes it difficult to achieve a balance between efficiency and accuracy in this class of methods. Alternatively, an iterative algorithm such as Newton’s method can be used for DOA estimation. In articles [15,16], the gradient descent algorithm and the Newton method are applied to DOA estimation with low complexity. In [17], the joint angle–frequency estimation problem of a multidimensional ESPRIT algorithm is discussed, and a novel Frame–Newton method is proposed to solve the singular value decomposition of complex asymmetric matrices based on the Newton method. The Newton method is applied to [16,18], and the former article uses the phase-comparison monopulse method to obtain the initial angle in Newton’s method and defines a cost function to find the peak for the MUSIC spectral function. The latter paper uses the Newton method as a rooting technique for global polynomials and specifies a minimum set of initial points for the Newton method using the Newton diagram of the Root-MUSIC polynomial to ensure that at least one of the initial points will converge to the range closest to the root of the unit circle. The two algorithms proposed above avoid falling into the local optimum in the gradient descent class of algorithms, but the estimation accuracy is slightly worse because they do not use the second-order derivatives in the Taylor expansion for the optimization calculation.

In this paper, a multidimensional joint parameter estimation algorithm for multistart patterns is proposed. The algorithm is based on the classical Newton method and extends the iterative optimization search to four-dimensional scales using Hessian matrices. An iterative optimization algorithm is proposed for multistart patterns with partitioned null fields in order to minimize the trapping in local optima and improve the accuracy of the first estimation. The threshold function of Newton’s method is proposed and improved to maximize the low number of iterations and the accuracy of the parameter estimation. In addition, we use the improved Nystrom method [19,20] to reduce the longitudinal dimension of the signal covariance matrix, which further reduces the computational burden of the algorithm. Simulation results show that the proposed algorithm takes less computation time than the conventional algorithm under the condition of guaranteed accuracy, and the experimental results further verify its performance.

The following is the structure of this paper. Section 2 introduces the array model used and its principles. Section 3 introduces the multidimensional joint estimation algorithm based on the Newton iteration method. Section 4 presents the detailed implementation process of the joint estimation algorithm for the multistart model, which includes the compression of the signal covariance matrix by the Nystrom method, the derivation of the multiorder derivatives and the iterative process under the corresponding array model, and the improvement of the threshold discriminator. The reliability and feasibility of the algorithm are verified by comparative experiments in Section 5.

## 2. Signal Model

Consider a scenario where *K* narrowband signals impinge upon an EM sensor array with an arbitrary manifold from far field, where the signals are fully polarized and mutually uncorrelated. To simplify the model analysis, the array are supposed to consist of *M* tripole sensors, which collect the electrical field components of impinging signals. The whole model is described in Figure 1.

The DOA and polarization parameters of the *k*th incident signal can be expressed as $\theta_{k}$, $\phi_{k}$, $\gamma_{k}$, $\eta_{k}$, *k* = 1, 2, …, *K*, where $\theta_{k}$, $\phi_{k}$∈ [0, π/2], $\gamma_{k}$, $\eta_{k}$∈ [0, 2π]. Interference noise is Gaussian white noise distributed with 0 mean and variance $\sigma_{n}^{2}$, which is uncorrelated with the incident signals. The array elements are irregularly distributed, and their position coordinates are $\left(x_{m},y_{m}\right)$, where m=1,2,…,M. The source signals impinge the different sensors with a time delay, which leads to the phase difference in received signals. Usually, the phase difference can be expressed with the steering vector $a_{k}$.
(1)$$a_{k}={\left[1,\ldots ,e^{-j\pi \left(x_{m}\cdot {\cos\theta }_{k}{\sin\phi }_{k}+y_{m}\cdot {\sin\theta }_{k}{\sin\phi }_{k}\right)/\lambda }\right]}^{T}$$
where $\theta_{k}$, $\phi_{k}$ are azimuth and elevation angles of the $k_{th}$ signal, and λ is the wavelength of the signal. According to [10], each EM sensor receives the signal in a 3D vector, which is influenced by the polarization vector $p_{k}$
(2)$$\begin{array}{c} \begin{array}{cc} p_{k} & =\left[\begin{array}{cc} {\cos\theta }_{k}{\cos\phi }_{k} & -{\sin\phi }_{k}\\ -{\cos\theta }_{k}{\sin\phi }_{k} & {\cos\phi }_{k}\\ -{\sin\theta }_{k} & 0 \end{array}\right]\cdot \left[\begin{array}{c} \sin\gamma_{k}e^{j\eta_{k}}\\ \cos\gamma_{k} \end{array}\right]\\ \; & =q_{k}g_{k} \end{array} \end{array}$$
where $q_{k}$ is only composed of DOA parameters, and $g_{k}$ includes the auxiliary polarization angle $\gamma_{k}$ and the polarization phase difference $\eta_{k}$

The received signal J(t) can be denoted as an algebraic sum of the scalar steering vector $a_{k}$, the polarization vector $p_{k}$, incident signal $s_{k}$, and the Gaussian white noise $n_{k}$ under a certain number of snapshots *L*, where
(3)$$J\left(t\right)=\sum_{k=1}^{K}[a_{k}⊗p_{k}]s_{t}+n_{t}=\sum_{k=1}^{K}z_{k}s_{t}+n_{t}$$
where ⊗ is the Kronecker product, $z_{k}$ stands for the Kronecker product of $a_{k}$ and $p_{k}$, and the Gaussian white noise $n_{k}$ is a 3M∗L vector. Regarding the covariance matrix of the received signal J(t), it can be expressed in terms of mathematical expectations
(4)$$R=E\left\{J\left(t\right)J{\left(t\right)}^{H}\right\}$$

In practical engineering applications, the covariance matrix is represented by taking the average of multiple snapshots.
(5)$$\hat{R}=\sum_{l=1}^{L}J\left(t\right)J{\left(t\right)}^{H}$$

## 3. The Proposed Algorithm

The traditional MUSIC algorithm applies an on-grid search to estimate the DOA and polarization, which loses estimation accuracy and brings massive computation complexity. In this section, an off-gird algorithm is proposed based on the multidimensional MUSIC algorithm. After obtaining the multivariate spectral function, Newton’s method is applied to find the extreme values. During the process, the first- and second-order derivatives regarding θ, ϕ, γ, and η are obtained by the spectral function. Then, a Hessian matrix is constructed to find out the extreme values, which locate the DOA and polarization results of the source signals. In order to avoid false results, a threshold determinant is applied at the end of the algorithm.

### 3.1. Processing Signals with the Nystrom Method

According to Equation (Equation 5), we can model the signal in a polarized steering vector. However, the dimension of the obtained covariance matrix is expanded to three times its original size compared to the traditional model. This directly leads to an increase in the complexity of the algorithm and a slower operation speed. Therefore, in this paper, the Nystrom method is utilized to chunk the received signal model and reconstruct the signal covariance matrix.

The received signal J is chunked and partitioned into the first 3K columns $J_{1}$ and the last M−3K columns $J_{2}$ according to the number of sources
(6)$$J=[J_{1},J_{2}]$$
and we obtain their respective subcovariance matrices $R_{ij}$
(7)$$\begin{array}{cc} \; & R_{11}=J_{1}\cdot J_{1}^{H}/L\\ \; & R_{12}=J_{1}\cdot J_{2}^{H}/L\\ \; & R_{21}=J_{2}\cdot J_{1}^{H}/L\\ \; & R_{22}=J_{2}\cdot J_{2}^{H}/L \end{array}$$

Finally, the signal covariance matrix after rank reduction using Nystrom’s method can be obtained by reconstructing R using the subcovariance matrix $R_{b}$.
(8)$$\begin{array}{c} R_{b}=R_{22}+R_{21}\cdot R_{1}^{-1}R_{12} \end{array}$$

In the matrix reconstruction using this method, the information of submatrix $R_{12}$ is not utilized, so the method saves some of the information used, and the dimension of the signal covariance matrix is reduced.

### 3.2. Rough Estimation

Before starting the rough estimation, we first use the singular value decomposition to obtain the noise subspace $U_{n}$ of the signal
(9)$$U_{space}\overset{svd}{\to }\left[U_{s},U_{n}\right]$$
and use the steering vector $z_{k}$ of the signal and the noise subspace $U_{n}$ to obtain the spectral peak function *f*
(10)$$f=z_{k}^{H}U_{n}U_{n}^{H}z_{k}=z_{k}^{H}V_{m}z_{k}$$

The rough estimation of the algorithm is then performed using the first-order Newton method. In this process, we need to use the original function of the spectral peak function *f* and its first-order derivative $df/d\theta_{k}$ and $df/d\phi_{k}$. We can obtain the gradient of *f* for the four components’ ($\theta_{k}$$\phi_{k}$$\gamma_{k}$$\eta_{k}$) directions according to the matrix derivative rule. It is worth noting that due to the different coupling relationships between the four parameters, the steering vector of the scalar array and the DOA component participate in the derivative operation as a whole ($\theta_{k}$ and $\phi_{k}$ are present in DOA components $q_{k}$, $\gamma_{k}$, and the $\eta_{k}$’s are present in polarization component $g_{k}$).
(11)$$\frac{df}{d\theta_{k}}=V_{m}^{T}z_{m}^{*}\frac{\partial q_{k}}{\partial \theta_{k}}g_{k}+V_{m}z_{m}{\left(g_{k}^{H}\frac{\partial q_{k}^{H}}{\partial \theta_{k}}\right)}^{T}$$
(12)$$\frac{df}{d\gamma_{k}}=V_{m}^{T}z_{m}^{*}q_{k}\frac{\partial g_{k}}{\partial \gamma_{k}}+V_{m}z_{m}{\left(\frac{\partial g_{k}^{H}}{\partial \gamma_{k}}q_{k}^{H}\right)}^{T}$$

Similarly, we can find the first-order derivative in the direction of $\phi_{k}$ and $\eta_{k}$ by simply replacing $\theta_{k}$ and $\gamma_{k}$. Usually, first-order derivatives are represented and operated on in matrix form $F_{1}=\left[\frac{df}{d\theta_{k}},\frac{df}{d\phi_{k}},\frac{df}{d\gamma_{k}},\frac{df}{d\eta_{k}}\right]$.

After several iterations of the first-order Newton method, a rough estimation can be described as
(13)$$x_{b}=x_{a}-\frac{f}{F_{1}}$$
where $x_{a}$ represents the initial position before the iteration, and $x_{b}$ represents the optimization result after the iteration in a single iteration. Please see the detailed explanation of the algorithm flow about Newton’s method in the Appendix A.

### 3.3. Accurate Estimation

The rough estimation is used as the initial point and the accurate estimation of the algorithm is performed using the second-order Newton method. However, unlike the rough estimation in this process, it is necessary to iterate using the first-order derivative matrix $F_{1}$ of the spectral peak function *f* and its second-order derivative. In Equations (14) and (15), we list in detail how to solve the second-order derivative equation based on the first-order derivative equation. There are four different solutions depending on the first-order derivative, which are the DOA components to polarization components for the partial derivatives, the DOA components to DOA components for the partial derivatives, the polarization components to polarization components for the partial derivatives, and the polarization components to DOA components for the partial derivatives.
(14)$$\begin{array}{cc} \frac{\partial^{2}f}{\partial \theta_{1}\partial \gamma_{1}}= & V_{m}^{T}g_{k}^{*}\frac{\partial q_{k}^{*}}{\partial \gamma_{1}}\frac{\partial q_{k}}{\partial \theta_{1}}g_{k}+V_{m}^{T}z_{k}^{*}\frac{\partial^{1}q_{k}}{\partial \theta_{1}\partial \gamma_{1}}g_{k}+\\ \; & V_{m}\frac{\partial q_{k}}{\partial \gamma_{1}}g_{k}\frac{\partial q_{k}^{*}}{\partial \theta_{1}}g_{k}^{*}+V_{m}z_{m}\frac{\partial^{1}q_{k}^{*}}{\partial \theta_{1}\partial \gamma_{1}}g_{k}^{*} \end{array}$$
(15)$$\begin{array}{cc} \frac{\partial^{2}f}{\partial \phi_{1}\partial \gamma_{1}}= & V_{m}^{T}(\frac{\partial q_{k}}{\partial \gamma_{1}}{g)}^{*}q_{k}\frac{\partial g_{k}}{\partial \phi_{1}}+V_{m}^{T}z_{m}^{*}\frac{\partial q_{k}}{\partial \gamma_{1}}\frac{\partial g_{k}}{\partial \phi_{1}}+\\ \; & V_{m}\frac{\partial q_{k}}{\partial \gamma_{1}}g{\left(\frac{\partial g_{k}^{H}}{\partial \phi_{1}}q_{k}^{H}\right)}^{T}+V_{m}z_{m}{\left(\frac{\partial g_{k}^{H}}{\partial \phi_{1}}\frac{\partial q_{k}^{H}}{\partial \gamma_{1}}\right)}^{T} \end{array}$$
(16)$$\begin{array}{cc} \frac{\partial^{2}f}{\partial \gamma_{1}\partial \eta_{1}}= & V_{m}^{T}(q_{k}\frac{\partial g_{k}}{\partial \eta_{1}}{)}^{*}q_{k}\frac{\partial g_{k}}{\partial \gamma_{1}}+V_{m}^{T}z_{m}^{*}q_{k}\frac{\partial^{2}g_{k}}{\partial \phi \gamma_{1}\eta_{1}}+\\ \; & V_{m}q_{k}\frac{\partial g_{k}}{\partial \eta_{1}}{\left(\frac{\partial g_{k}^{H}}{\partial \gamma_{1}}q_{k}^{H}\right)}^{T}+V_{m}z_{m}{\left(\frac{\partial g_{k}^{H}}{\partial \gamma_{1}\eta_{1}}q_{k}^{H}\right)}^{T} \end{array}$$

Similarly, we can find the second-order derivative in all the directions by simply swapping $\theta_{k}$ with $\phi_{k}$ and $\gamma_{k}$ with $\eta_{k}$ according to Equation (Equation 16).

In the algorithm proposed in this paper, the given objective spectral function is complex and requires a large number of dimensions for the joint estimation. In order to make the four directional parameters keep the synchronization during an iteration, the Hessian matrix is introduced to simplify the problem in Equation (Equation 17). The Hessian matrix was introduced in the 19th century by the German mathematician Ludwig Otto Hesse, and it is a diagonal array of second-order partial derivatives of multivariate functions that can be used to describe the local curvature of a function. When a multivariate real function $f(x_{1},x_{2},\ldots ,x_{n})$ has second-order continuous partial derivatives in the neighborhood of the point $x_{0}$, then its Hessian matrix at the point $x_{0}$ can be described as
(17)$$H\left(x_{0}\right)=\left[\begin{array}{cccc} \frac{\partial^{2}f}{\partial x_{1}^{2}} & \frac{\partial^{2}f}{\partial x_{1}\partial x_{2}} & \ldots \; & \frac{\partial^{2}f}{\partial x_{1}\partial x_{n}}\\ \frac{\partial^{2}f}{\partial x_{2}\partial x_{1}} & \frac{\partial^{2}f}{\partial x_{2}^{2}} & \ldots \; & \frac{\partial^{2}f}{\partial x_{2}\partial x_{n}}\\ \vdots & \vdots & \ddots & \vdots \\ \frac{\partial^{2}f}{\partial x_{n}\partial x_{1}} & \frac{\partial^{2}f}{\partial x_{n}\partial x_{2}} & \ldots \; & \frac{\partial^{2}f}{\partial x_{n}^{2}} \end{array}\right]$$

The Hessian matrix consists of a set of second-order partial derivatives corresponding to each variable. Using the Hessian matrix, sufficient conditions for the determination of extreme-value points of multivariate functions can be summarized as
H (positive definite) ⇔ the function $f(x_{1},x_{2},\ldots ,x_{n})$ has a minimal value at point $x_{0}$.H (negative positive definite) ⇔ the function $f(x_{1},x_{2},\ldots ,x_{n})$ has a maximum value at point $x_{0}$.H (nonpositive definite) ⇔ the function $f(x_{1},x_{2},\ldots ,x_{n})$ has no extreme value point at point $x_{0}$.

After several iterations of the second-order Newton method, an accurate estimation can be described as
(18)$$x_{c}=x_{b}-\frac{F_{1}}{F_{2}}$$
where second-order derivatives are represented and operated on in matrix form $F_{2}$, $x_{b}$ is the result of the rough estimation, and $x_{b}$ is the result of the accurate estimation after many iterations, where $F_{1}$ has the form of a vector, and $F_{2}$ has the form of a matrix
(19)$$F_{1}=\left[\frac{df}{d\theta_{k}},\frac{df}{d\phi_{k}},\frac{df}{d\gamma_{k}},\frac{df}{d\eta_{k}}\right]$$
(20)$$F_{2}=\left[\begin{array}{cccc} \frac{\partial^{2}f}{\partial \theta_{k}^{2}} & \frac{\partial^{2}f}{\partial \theta_{k}\partial \phi_{k}} & \frac{\partial^{2}f}{\partial \theta_{k}\partial \gamma_{k}} & \frac{\partial^{2}f}{\partial \theta_{k}\partial \eta_{k}}\\ \frac{\partial^{2}f}{\partial \phi_{k}\partial \theta_{k}} & \frac{\partial^{2}f}{\partial \phi_{k}^{2}} & \frac{\partial^{2}f}{\partial \phi_{k}\partial \gamma_{k}} & \frac{\partial^{2}f}{\partial \phi_{k}\partial \eta_{k}}\\ \frac{\partial^{2}f}{\partial \gamma_{k}\partial \theta_{k}} & \frac{\partial^{2}f}{\partial \gamma_{k}\partial \phi_{k}} & \frac{\partial^{2}f}{\partial \gamma_{k}^{2}} & \frac{\partial^{2}f}{\partial \gamma_{k}\partial \eta_{k}}\\ \frac{\partial^{2}f}{\partial \eta_{k}\partial \theta_{k}} & \frac{\partial^{2}f}{\partial \eta_{k}\partial \phi_{k}} & \frac{\partial^{2}f}{\partial \eta_{k}\partial \gamma_{k}} & \frac{\partial^{2}f}{\partial \eta_{k}^{2}} \end{array}\right]$$

### 3.4. Threshold Determiner

In order to maximize the estimation accuracy while reducing the number of iterations of the algorithm, the threshold of the algorithm is improved in this paper. Unlike the traditional Newtonian thresholding discriminant function, we divide the original thresholding function into two parts. One part is used as the discriminant function of the global meritocratic algorithm, and the other part is used as the discriminant function of the basic meritocratic algorithm. Moreover, a function for controlling the algorithm cycle is added for each of them, and the search is reperformed if the current search result does not satisfy the minimum threshold conditions of the spectral function, gradient, and Hessian matrix. The following is the threshold discriminant step of the two-part optimization seeking algorithm. The threshold discriminator for the global optimization algorithm and basic optimization algorithm are shown in Figure 2 and Figure 3.

The main function of the global discriminator in Figure 2 is to filter out a preliminary estimate that is close to the target to be measured. It is mainly judged by the height of the spectral peaks and whether or not the extreme points are reached. The process may be repeated three to five times until an ideal preliminary estimate is found.

When the initial set of random input points $x_{b}$ is not ideal, it may cause the global optimization result $x_{c}$ to fall into a local optimum (optimization results are considered to be trapped in a local optimum if the spectral function value at that point is nonzero). In this case, the discriminator will then repick the value of the initial set of points $x_{b}$ and optimize again until the estimated result is ideal.

When the signal-to-noise ratio is low, the convergence of the basic optimization algorithm becomes difficult because the effect of the Gaussian white noise on the signal becomes larger. When the number of iterations reaches a preset maximum (i.e., it fails to converge within the preset number), the discriminator adjusts the global optimization result in a very small neighborhood ($x_{c}$, $x_{c}$ + $\Delta x_{c}$) and performs a new basic optimization again. This process is repeated until the desired result is obtained.

### 3.5. Algorithm Steps

The proposed algorithm in this paper can be roughly divided into four parts: the data-processing part, the global optimization part of the multistart model, the basic optimization part, and the discriminator part. The detailed steps of the algorithm are as follows.

Step 1: Obtain the received signal J(k) by Equation (Equation 5), chunk the matrix, and reconstruct the new covariance matrix $R_{b}$. Then, obtain the noise subspace $U_{n}$ by Equation (Equation 9).

Step 2: Constructing the target spectral function iteratively for Newton’s method, the expression of the spectral function *f* can be obtained according to Equation (Equation 10)

Step 3: Use Equations (11)–(17) to find the gradient $F_{1}$ and the Hessian matrix $F_{2}$ of *f* in the four directions of $\theta_{k}$, $\phi_{k}$, $\gamma_{k}$, and $\eta_{k}$

Step 4: Divide the neighborhood of the space equally into 9 parts, i.e., spaces 1 to 9, as shown in Figure 4.

A starting point $x_{b}$ = [$x_{b1}$, $x_{b2}$, …, $x_{b9}$] is randomly taken within each group of empty space. To ensure that the starting points put in the space can cover the whole empty space, try to keep the points in the adjacent empty space at a certain interval. The global optimization algorithm of the multistart pattern is performed in nine spaces. It is worth noting that in order to keep the four angle components [$\theta_{k}$, $\phi_{k}$, $\gamma_{k}$, $\eta_{k}$] highly iterative and synchronous during the global optimization, the first-order derivative matrix and the second-order derivative matrix should be involved in the form of vectors $F_{1}$ and matrices $F_{2}$.

Update the starting point using the global optimization algorithm until convergence.

Step 5: The obtained estimations $x_{c}$ are used as a starting point for the basic optimization algorithm and are further optimized to obtain higher-accuracy estimations.

Step 6: A judgment is performed on $x_{c}$ and $x_{d}$ using the adjudicators in Figure 2 and Figure 3. When the judgment is finished, $x_{d}$ is the estimated parameter. The DOA component is [θ, ϕ] = [$x_{d}$(1), $x_{d}$(2)], and the polarization component is [γ, η] = [$x_{d}$(3), $x_{d}$(4)], where $x_{d}\left(m\right)$ refers to the *m*th element of the matrix $x_{d}$.

## 4. Simulation Results

In order to show more clearly the efficiency of the proposed algorithm, we conducted experiments using Monte Carlo experiments. It is worth stating that in the experiments, in order to more conveniently and quickly reflect the advantages that the algorithm can work in any array manifold and to demonstrate the performance of the proposed algorithm in this paper, the experiments were all conducted under uniform linear arrays. In the experiments, we assumed that there were 10 tripole sensor arrays. Two uncorrelated far-field narrowband signals were impinging into the array, and *L* = 300 snapshots were collected (the term “preconditioned” is used in the following to refer to “*M* = 10, *K* = 2, *L* = 300”). All simulations were performed in MATLAB R2021a. The RMSE is defined as
(21)$$RMSE=\frac{1}{K}\sum_{k=1}^{K}\sqrt{\frac{1}{L}\sum_{l=1}^{L}{\left(P_{l,k}-P_{k}\right)}^{2}}$$
where *L* is the total number of experiments, $P_{k}$ and $P_{l,k}$ are the true and measured values of the *k*th experiment.

In the first experiment, we conducted 300 Monte Carlo experiments on the proposed algorithm for different SNRs in the preconditioned state, observed the accuracy of the algorithm, and recorded its RMSE. The results show that the algorithm has good performance in solving the multidimensional joint parameter estimation problem and can be used in practical engineering. Figure 5 and Figure 6 are the pictures of error analysis of two directions. These two figures present the error between the estimated results and the true values, i.e., the RMSE analysis plots of the angles, for the two sets of target angles goal1 and goal2. There are four sets of data for each target angle, which are azimuth angle, elevation angle, auxiliary polarization angle, and the polarization phase difference.

In this paper, the proposed method can be applied to arrays of any shape, and compared to other algorithms where the array manifold must satisfy certain conditions (e.g., the trilinear decomposition in the propagator method must ensure that the array manifold can provide information in multiple dimensions), the application scenario is much broader. However, the estimation of the four parameters ($\theta_{k}$, $\phi_{k}$, $\gamma_{k}$, $\eta_{k}$) can be performed only using a uniform linear array, in the case where trilinear dipole array elements are used.

In addition, the effectiveness of the basic optimization algorithm process was verified by building noise with different structures and comparing the results. First, an ideal noise $n_{k}$ with a diagonal array structure was added to a source $s_{k}$ in the ideal case, and the results of the global optimization algorithm and the basic optimization algorithm were compared in the same estimation process (when the noise is ideal enough, the estimation error of the latter should be infinitely close to zero, and the estimation accuracy of the former is slightly worse). Figure 7 shows the comparison of the global Newton method and the basic Newton method errors for each SNR in the ideal noise case, which confirms the validity of the basic Newton method.

### 4.1. RMSE Comparison

In the second experiment, the proposed method was used to compare the performance with various classical subspace algorithms, where the search range was [−90°, 90°], and the search step was 0.1° in the preconditions. In order to represent the simulation results more intuitively, we represented the RMSE in the form of a two-parameter number in the experiments comparing it with that of other algorithms in the preconditions (i.e., the straight-line distance between the true value and the measured value in the spatial domain). It can be seen in Figure 8. The spectral peak search algorithm is limited by the grid density leading to its own performance limitation at the 0.1° step. In contrast, the proposed method is an iterative algorithm with a variable step size, which has a very high convergence accuracy. This experiment was conducted at 5 dB–25 dB, a SNR that is more common and prevalent in everyday life. When the SNR is higher, the performance of the four algorithms will be closer.

In the third experiment, a comparison of the two RMSEs of the proposed algorithm before and after utilizing the rank reduction method was conducted. In Figure 9, it can be seen that the covariance matrix is less affected by the accuracy of the algorithm after the rank reduction. Almost the full accuracy of the algorithm is retained.

### 4.2. Time Comparison

In the fourth experiment, we performed a statistical comparison of the average time for a single run of each algorithm, as shown in Figure 10. The spectral peak search algorithm improves the algorithm performance at the expense of time in order to ensure its own excellent estimation accuracy. Moreover, it continues to increase the computation time as the grid is refined. In the proposed method, in addition to the advantage of variable step size, the presence of the adjudicator further saves the algorithm a lot of time cost. It is worth mentioning that the average computation time of the proposed method is more irregular for different SNRs than other algorithms. The reason is that we cannot guarantee that the adjudicator termination threshold is at the same level for each SNR (i.e., it is possible that if the threshold is slightly higher, at SNR = 10, the adjudicator will increase the operation time in order to get a more accurate estimation).

In the fifth trial, The efficiency of the improved Nystrom method for the rank reduction was verified by comparing the average time of a single run of the proposed algorithm before and after the same condition of rank reduction, as shown in Figure 11. In combination with the above analysis, it can be seen that our method not only retains the accuracy of the original algorithm to a great extent but also reduces the time used for a single computation of the algorithm significantly.

## 5. Conclusions

In this article, we proposed a multidimensional joint parameter estimation algorithm. The algorithm took the iterative optimization algorithm as the core, extended the joint estimation algorithm to 4D dimensions using the Hessian matrix, and a new threshold judge was constructed based on the spectral function and gradient, which achieved effective control over the iterative process of the algorithm. The proposed algorithm overcame the dependence of traditional subspace algorithms on the spectral peak search and dimensionality reduction methods. The algorithm achieved a very low number of iterations while ensuring strong convergence, balancing a high estimated accuracy and a low algorithmic complexity. The simulation experiments confirmed our theoretical analysis. Therefore, it is of practical significance for improving array signal processing theory and multidimensional joint parameter estimation.

## Figures and Tables

**Figure 1 sensors-23-08193-f001:**
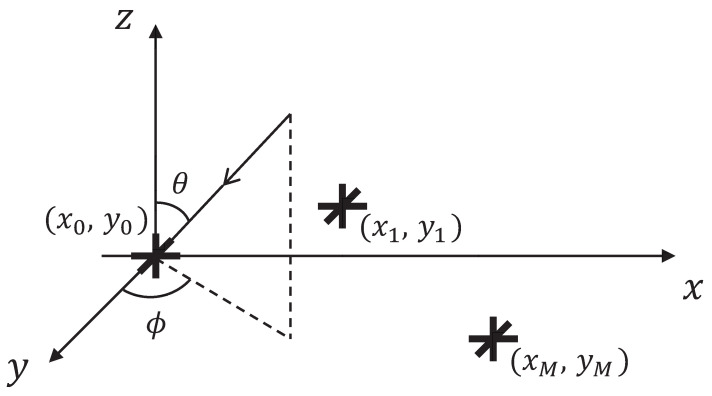
An arbitrary manifold.

**Figure 2 sensors-23-08193-f002:**
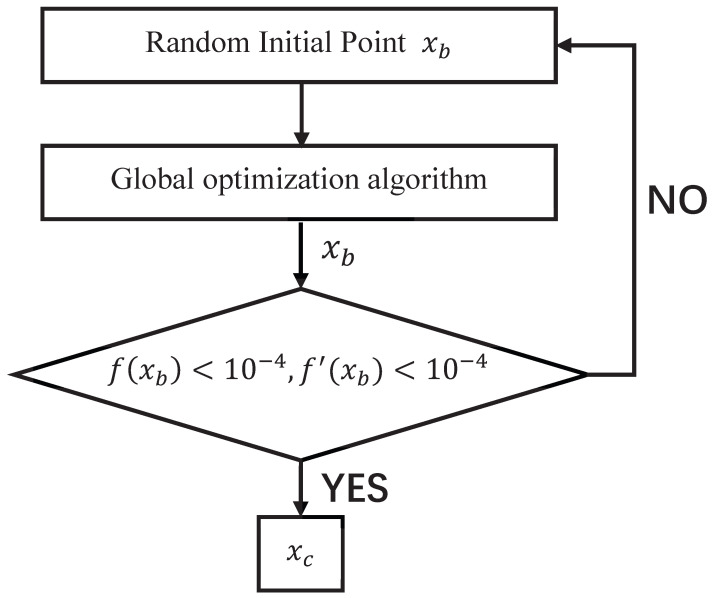
Global Discriminator.

**Figure 3 sensors-23-08193-f003:**
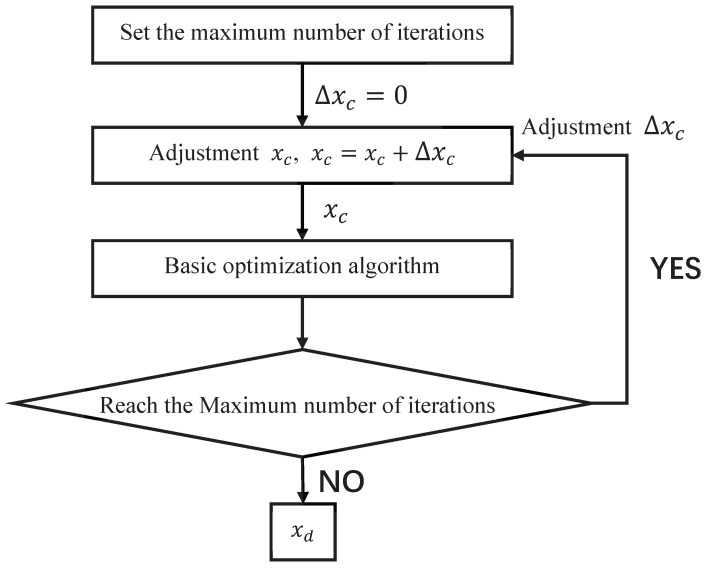
Basic Discriminator.

**Figure 4 sensors-23-08193-f004:**
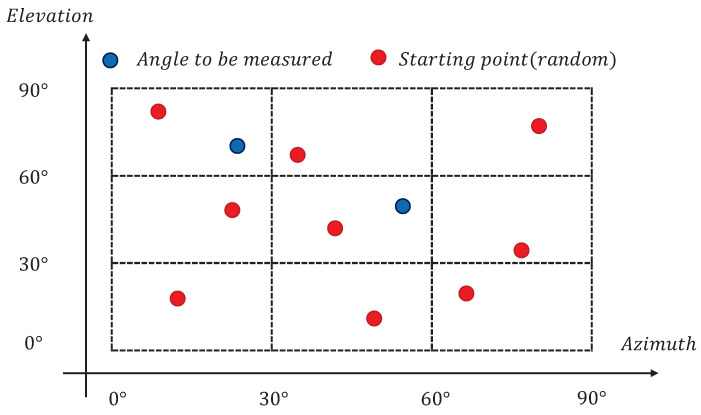
Airspace division diagram.

**Figure 5 sensors-23-08193-f005:**
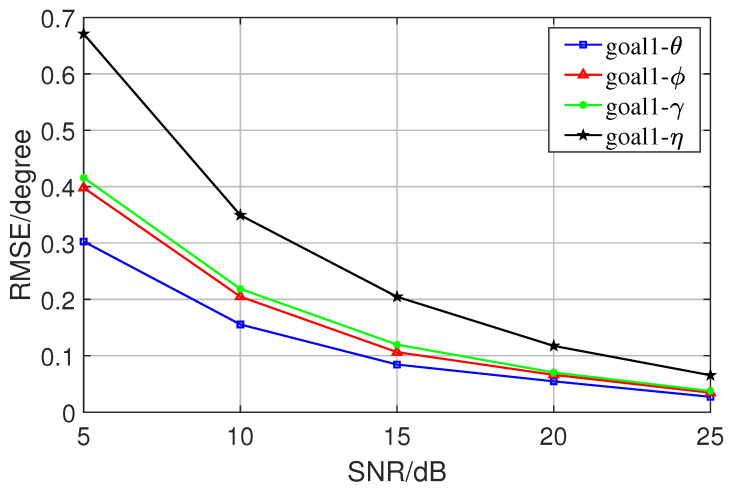
RMSE of the first goal.

**Figure 6 sensors-23-08193-f006:**
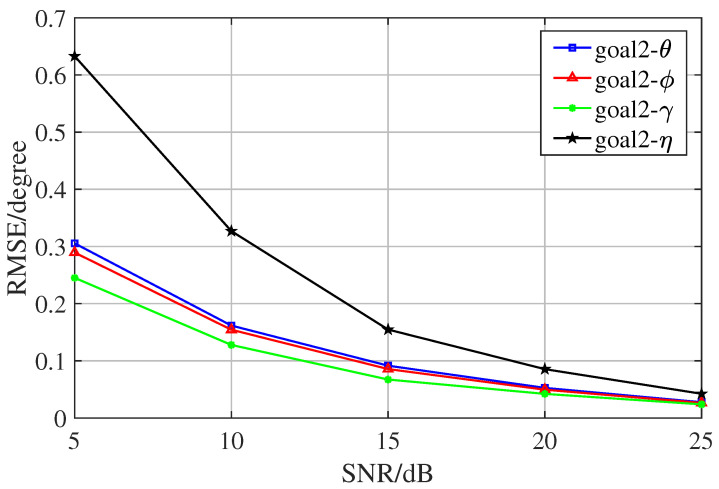
RMSE of the second goal.

**Figure 7 sensors-23-08193-f007:**
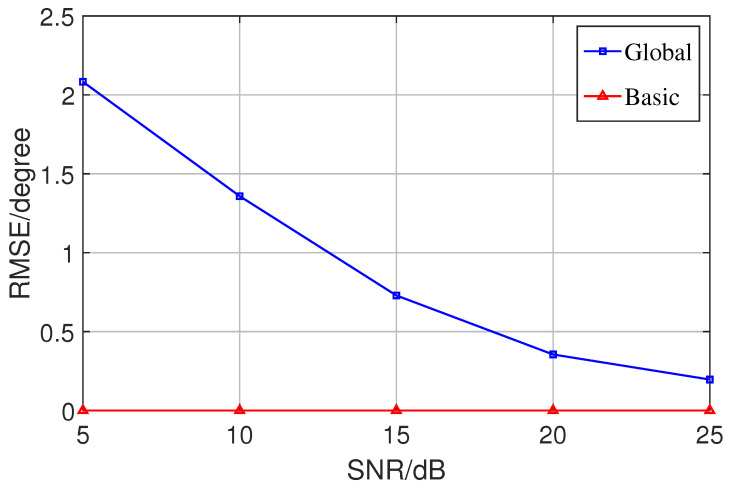
Comparison of the effectiveness of basic optimization algorithms under ideal signal and noise conditions.

**Figure 8 sensors-23-08193-f008:**
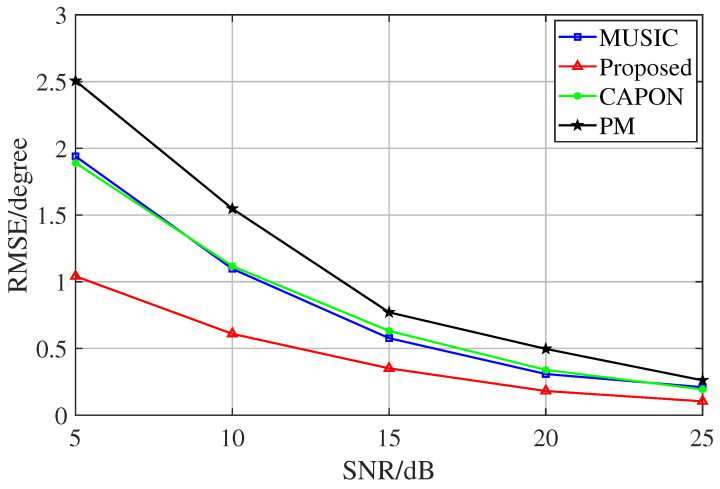
RMSE comparison of different algorithms.

**Figure 9 sensors-23-08193-f009:**
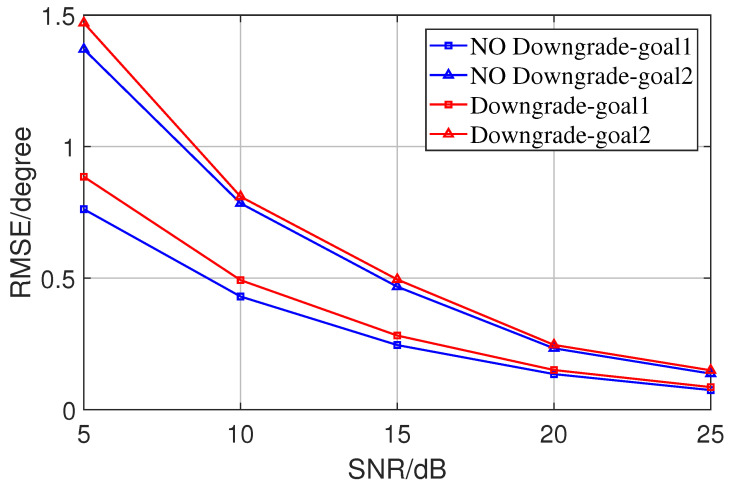
Comparison of the RMSE before and after rank reduction.

**Figure 10 sensors-23-08193-f010:**
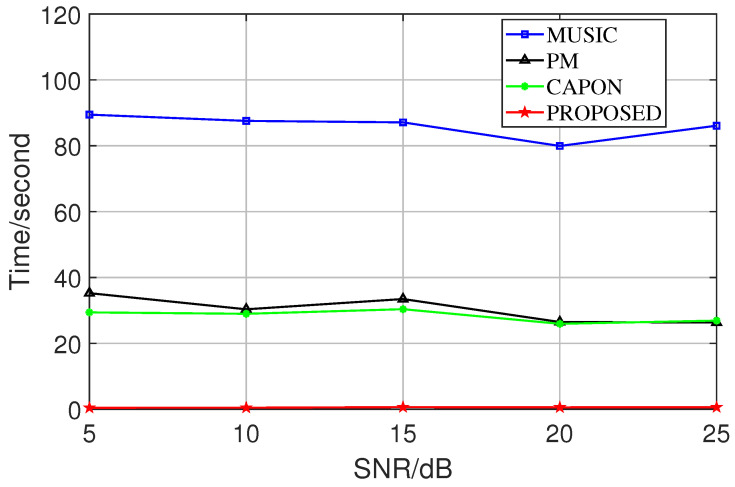
Time comparison of different algorithms.

**Figure 11 sensors-23-08193-f011:**
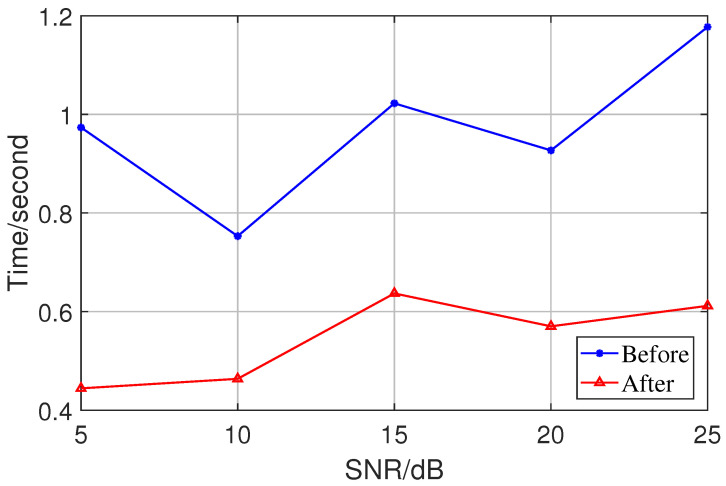
Comparison of time before and after rank reduction.

## Data Availability

No applicable.

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
