# Peer review of "A Fast Method for Multidimensional Joint Parameter Estimation of Polarization-Sensitive Arrays"

_sensors, 2023, doi:10.3390/s23198193_

Round 1

Reviewer 1 Report

The paper proposes a fast method for multidimensional parameter estimation of polarization sensitive array. It is an interesting topic, and it fits in the journal scope. However, there are a few parts I would like the writers to introduce it more clearly.

1. I don't understand the Figs. 5 and 6. What are the meanings for the four lines?

2. For Fig. 8, you compared the RMSE value, what will it be like when the SNR is over 25dB? Will the lines be flatted over a certain value of SNR?

The English is OK

Reviewer 2 Report

1. The authors propose a fast method for multidimensional parameter estimation of polarization sensitive array. Compared with conventional methods (e.g. MUSIC algorithm), the proposed method applies an iterative approach based on Newton method to obtain joint estimation results

instead of spectral search and dimension reduction.

2.      Please elaborate the concept of the equations 15, 16, 17 and 18 in detail.

3.      In the figure 2, global discriminator, should be elaborated in detail.

4. The authors propose a fast method for multidimensional parameter estimation of polarization sensitive array.  Compared with conventional methods (e.g. MUSIC algorithm), the proposed method applies an iterative approach based on Newton method to obtain joint estimation results instead of spectral search and dimension reduction.

5. The authors should emphasize contribution and novelty, the introduction needs to clarify the motivation, challenges, contribution, objectives, and significance/implication. 

6. Please compare the contributions of the proposed multidimensional parameter estimation technology to related technologies, in detail.

7. Please elaborate the technique efficacy of the proposed multidimensional parameter estimation method in detail.

8. Please develop the concept of the equations 15, 16, 17 and 18 in detail.

9. In the figure 2, global discriminator, should be dilated in detail.

10. Please thoroughly check the language form before the  submission article.

Reviewer 3 Report

Thanks for your good efforts about this paper!

This paper is very interesting in Polarization Sensitive Array.

I believe that this paper will be good motivation about this field.

Thanks again!
